# Talking about Climate Change and Environmental Degradation with Patients in Primary Care: A Cross-Sectional Survey on Knowledge, Potential Domains of Action and Points of View of General Practitioners

**DOI:** 10.3390/ijerph19084901

**Published:** 2022-04-18

**Authors:** Hélène André, Julia Gonzalez Holguera, Anneliese Depoux, Jérôme Pasquier, Dagmar M. Haller, Pierre-Yves Rodondi, Joëlle Schwarz, Nicolas Senn

**Affiliations:** 1Department of Family Medicine, Center for Primary Care and Public Health (Unisanté), University of Lausanne, 1004 Lausanne, Switzerland; joelle.schwarz@unisante.ch (J.S.); nicolas.senn@unisante.ch (N.S.); 2Competence Center for Sustainability, University of Lausanne, 1015 Lausanne, Switzerland; julia.gonzalezholguera@unil.ch; 3Centre Virchow-Villermé and Centre des Politiques de la Terre, Université Paris Cité, 75006 Paris, France; anneliese.depoux@uspc.fr; 4Sector of Biostatistics, Center for Primary Care and Public Health (Unisanté), University of Lausanne, 1004 Lausanne, Switzerland; jerome.pasquier@unisante.ch; 5University Institute for Primary Care, Faculty of Medicine, University of Geneva, 1206 Geneva, Switzerland; dagmar.haller-hester@unige.ch; 6Institute of Family Medicine, University of Fribourg, 1700 Fribourg, Switzerland; pierre-yves.rodondi@unifr.ch

**Keywords:** co-benefits, planetary health, general practice

## Abstract

Purpose: General practitioners (GPs) could play a role in mitigating climate change by raising awareness of its impact on human health and implementing changes to improve population health and decreasing environmental footprints. The aim of this study was to assess GPs’ knowledge and perspectives about the health impacts of climate change. Method: A questionnaire was sent to 1972 GPs in the French-speaking part of Switzerland. Knowledge of the impact of environmental degradations and climate change on health and willingness to address climate change with patients, to be exemplary and to act as role models were surveyed as well as demographic characteristics of GPs. Results: Respondents (N = 497) expressed a high level of self-reported knowledge regarding climate change, although it was lower for more specific topics, such as planetary health or health–environment co-benefits. Participants mostly agreed that it is necessary to adapt clinical practice to the health impacts of climate change and that they have a role in providing information on climate change and its links to human health. Conclusion: Most of the GPs were concerned about environmental and climate degradation. However, this study revealed a gap between the willingness of GPs to integrate the impact of climate change on health into their clinical activities and their lack of overall knowledge and scientific evidence on effective interventions. A promising way forward may be to develop co-benefit interventions adapted to the clinical setting on diet, active mobility and connecting with nature.

## 1. Introduction

The current situation of climate change and global environmental degradation is the greatest challenge of this century [1]. Recently, nine planetary boundaries were suggested for human-linked disruption of critical Earth processes, the thresholds of which should not be crossed on a global level to ensure sustainable living conditions [2]. Several of these boundaries, including climate change and biodiversity loss, have already been crossed as a result of human activity. This disruption of the Earth’s ecosystems is increasingly recognized as a critical public health issue. The Intergovernmental Panel on Climate Change (IPCC) reports, which describe the current trend of global warming and modeled projections, emphasizes the need to limit global warming to below 1.5 °C in order to ensure favorable living conditions [3]. All these statements stress that vulnerable populations, such as the elderly, children and deprived people, will be the most affected, resulting in increased health inequalities [4]. Clinical practice is already impacted and will be further affected by climate change, and thus clinicians will have to anticipate and adapt healthcare to the burden of diseases induced by climate change. Among the health impacts due to climate change, clinicians will have to address specific problems, such as heat-related diseases, vector-borne diseases or eco-anxiety [5,6]. They could also play a key role in disseminating accurate information on the health impact of climate change to patients, the population, colleagues or policymakers.

There is a growing awareness in the medical community about the threats posed by climate change to human health. A number of position papers highlight the moral and ethical duties of healthcare professionals to come up with an effective public health response [7,8,9], including disseminating accurate information on the health impact of climate change to patients and populations and advocating at the policy level for structural changes [10]. In this perspective, the new concept of “Planetary health” is increasing exponentially in healthcare academic circles.

This new outlook on health, which builds on environmental and Earth system science notions such as “ecosystem functioning” and “planetary boundaries”, is gaining popularity in the literature. Planetary boundaries, defined as thresholds of ecosystem transformations that should not be exceeded at the global level to ensure sustainable living conditions, have been identified for nine areas [11]. Many of these boundaries are either in a zone of concern or already beyond this zone as a result of anthropogenic environmental degradations. These disruptions of the Earth’s ecosystems are increasingly recognized as critical public health. Yet, it is unclear how and whether this viewpoint is effectively reaching practicing clinicians. In particular, we do not know to what extent clinicians recognize that environmental determinants of health are not limited to local exposure (e.g., air pollution, water quality), but extend to ecological functioning of the planet itself, providing favorable conditions for both humans and non-humans to thrive (for example, through climate regulation or pollination) [4,12]. Finally, if we understand planetary health as a global public health view, it is less clear how we could translate it into meaningful individual and community healthcare interventions.

A number of papers proposed that, with their knowledge and expertise in the management of diseases and ability to advocate health prevention and promotion, clinicians are well-positioned to boost behavior that provides benefit for both humans and environment health [9,13,14]. In that sense, the “co-benefit” approach, which considers how climate change mitigation action can also lead to health improvements, has been repeatedly highlighted [15,16]. This includes, for example, active mobility (favoring active transport modes instead of fossil-fueled modes) or a switch to diets based mostly on vegetables and whole grains as recommended by the Eat-Lancet commission [14,17,18]. Indeed, the WONCA (World Organization of National Colleges, Academies and Academic Associations of General Practitioners/Family Physicians.) adopted a declaration calling for family doctors around the world to act on planetary health [19], which briefly lists co-benefits for health and the environment that could be aimed at patients. However, there are currently no clear recommendations on how to address this notion with patients. Furthermore, it is even less clear how to coordinate meaningful individual interventions and health system policy change.

As a first step towards developing such interventions, this study aimed to determine the level of knowledge of general practitioners (GPs) regarding health impacts of climate change and of concepts such as planetary health and co-benefit interventions, as well as their willingness to address climate change with patients. In addition, we aimed to explore more generally the role that physicians are willing to play in society, as health advocates, for example. This study focused on GPs as they are the first point of contact of healthcare and arguably potential leaders in drawing the attention of communities to climate change and advocating change at the policy level due to their proximity to the community they serve [14,20]. These statements are reinforced by the fact that GPs are usually considered as one of the most trusted sources of information by patients, including for environmental health issues [21].

## 2. Method

### 2.1. Study Design

This is a cross-sectional survey among GPs in the French-speaking part of Switzerland conducted between January and June 2021. We developed a 40-question survey in French, based on existing questionnaires [22,23,24,25] and on published recommendations, such as the WONCA declaration [19]. The survey was pilot-tested with six GPs, one sociologist and one environmental scientist to ensure clarity, scientific accuracy and length. The final questionnaire took about 20 min to complete.

### 2.2. Contact Procedure

We contacted 1972 family physicians using postal mail and email lists of cantonal medical associations (this corresponds to approximately 80% of all practicing GPs in the region, as registration is not compulsory). Then, we sent out two reminders 2–3 weeks apart by e-mail or post, depending on the availability of email addresses.

Each initial invitation to participate in the survey included a paper format questionnaire and a website address to access the secured survey platform, REDCap.

### 2.3. Variables of Interest

We divided the survey into four sections, as below.

#### 2.3.1. Environmental Degradation Knowledge and the Impact on Health

We asked participants about their general knowledge of climate change, planetary health, planetary boundaries and health–environment co-benefits, health–environment concepts and the respective links with health using self-reported questions and objective questions. For self-reported questions, participants estimated their knowledge of the topic from 0 (no or little knowledge) to 100 (excellent knowledge). After each question, a brief definition of the concept was given, so they were able to progress through the questionnaire.

#### 2.3.2. Willingness to Address Climate Change with Patients

We assessed GPs’ willingness to address environmental issues with patients, and more specifically the health impact of climate change, asking GPs if: in their opinion, they had a role in providing information on this topic to patients and if they would feel comfortable advising patients on the topic. In addition, we asked what proportion of a medical consultation could be dedicated to climate change, according to them. Then, they were asked to rate, on a five-point Likert scale, how legitimate they felt discussing each of the seven co-benefit points from the WONCA declaration—food choices, mobility, energy choices, contact with nature (i.e., spending more time outside in nature could bring health benefits and increase a sense of stewardship for the natural environment), reproductive health (i.e., ensuring access to reproductive healthcare to improve women’s health and limit population growth by limiting unwanted pregnancies), reducing personal environmental impact in other ways and engaging in the community [19]. Finally, we asked them to identify barriers preventing them from discussing these topics and any resources they thought they would need to promote addressing climate change with patients.

#### 2.3.3. Exemplarity and Role Model

We assessed GPs’ willingness to take steps to personally reduce their own carbon footprint. Questions were based on the WONCA declaration [19].

#### 2.3.4. Demographic Characteristics

Finally, we collected general demographic data, including age, gender, practice setting and, in order to ensure the representativeness of our sample and characterize potential selection bias, overall political orientation.

## 3. Data Analysis

In the first place, we used standard descriptive analyses (i.e., frequencies and percentages for categorical variables and means and standard deviations (SD) for continuous variables) to summarize the study variables. Secondly, we performed basic bivariable analyses to explore how demographic characteristics influenced GPs’ answers concerning variables of interest. Finally, we used multiple linear and logistic regression models to estimate the association between variables of interest (6 binary predictive variables, listed in the first row of Table 3) and the main demographic characteristics, which were (age (continuous), gender (2 categories), place of practice (3 categories) and political orientation (3 categories). We used statistical software R 4.1.3 (R Core Team, Vienna, Austria). We considered “No opinion” responses or incongruent responses (i.e., indicating having no political affiliation as well as selecting a political affiliation) as missing data.

## 4. Results

A total of 514 GPs responded, and 17 questionnaires were discarded because of the poor quality of data (i.e., less than 50% of the questions were completed or demographic data missing), which left 497 questionnaires for analysis, with 171 hard copies and 326 in an electronic format (response rate of 25.2%). The mean age of responders was 52 years, 53% were women, and 21% declared being politically right-wing, 35% left-wing, and the others either declared no political affiliation or were not willing to declare it. The complete demographic characteristics are described in Table 1.

### 4.1. General Knowledge of Environmental Issues

GPs showed a high level of self-reported general knowledge regarding climate change, with over half of the respondents reporting having a basic knowledge of the issue. However, for more specific subjects, such as planetary boundaries, planetary health or co-benefits, self-reported knowledge was lower (Figure 1).

After reading the description of the planetary health concept, respondents were asked if this concept was of interest for clinical practice. Over half of the respondents (52%) agreed or mostly agreed, almost half had no opinion (43%) and a minority disagreed or strongly disagreed (5%).

### 4.2. Willingness to Address Climate Change with Patients

Participants mostly agreed (80%) that it is necessary to adapt clinical practice to the health impacts of climate change and that GPs have a role in providing information to patients about climate change and its links to human health. However, over half of the GPs admitted not being comfortable advising patients about the impact climate change could have on their health (Table 2). Regarding the topics they would readily address to encourage patients to reduce their environmental footprint, they mainly agreed on the subjects of food choices, transport means and time spent in nature, while they mainly disagreed with addressing reproductive health. Opinions were more balanced for energy choices, engaging in the community and reducing personal environmental impacts in other ways. (Figure 2).

Most of the participants (78%) reported that climate change is sometimes addressed during medical consultations. Of these, 44% declared that climate change is addressed in more than 10% of consultations. The physician or the patient brought up the topic equally as often during the consultation. Regarding co-benefits, 77% of survey respondents agreed or strongly agreed that this approach may help them speak about climate change with a patient.

Participants were asked to identify barriers preventing them from engaging in climate change discussions with patients. The majority said that lack of time was a barrier (70%), and also lack of clinical recommendations (62%) and lack of knowledge (57%). Eighteen percent of participants responded that patients would not be interested in this topic, 18% said that it is not the role of GPs to address this topic, 11% stated that discussion of this matter is not billable and 7% responded that there is no connection between climate change and health.

### 4.3. Training and Source of Information

A large majority of the respondents said that they were willing to learn more about climate change if given the opportunity (89%). Participants identified continued medical education (88%) and university lectures (86%) as potentially useful learning resources.

Respondents were asked to identify means that may be helpful to disseminate information to patients and the community. They showed approval for patient education material, such as flyers in waiting rooms (79%), policy statements provided by medical associations (81%), community advocacy (74%) and raising public awareness in the media (70%). Views on political involvement were mixed, with 41% of participants stating that they disagreed or strongly disagreed and 49% stating that they agreed or strongly agreed, and 10% had no opinions.

#### 4.3.1. Exemplarity and Role Models

Most of the respondents thought that GPs could serve as role models for the population in terms of sustainability (78% agreed or strongly agreed). They mainly reported being willing to make choices in their day-to-day life to lessen their own carbon footprint (97% agreed or strongly agreed). Regarding co-benefit items from the WONCA declaration, findings were similar to the responses concerning willingness to address themes with patients. They mainly agreed with all the proposed actions, except for addressing reproductive health in the context of limiting population growth and, to a lesser extent, engaging themselves in their communities (Figure 2).

#### 4.3.2. Influence of Demographic Factors on Respondents’ Answers

As we expected, answers were strongly and significantly associated with GPs’ political affiliation and age, and to a lesser extent, gender. Globally, female respondents reported having significantly less knowledge of climate change (excellent knowledge: 37% in females vs. 48% in males, *p* < 0.001). Regarding addressing climate change with patients, older GPs reported feeling comfortable addressing topics with their patients (*p* < 0.001), addressed climate change matters more often during medical visits (*p* < 0.001), were more willing to integrate climate change into the clinical setting (*p* = 0.002), and were more of the opinion that GPs could provide information to patients about the health impact of climate change (*p* = 0.01) and could be role models in helping to limit carbon footprints (*p* < 0.001).

Finally, political orientation was an important variable as GPs who declared having a left-wing political orientation were more knowledgeable on climate change topics (*p* < 0.001), more willing to integrate climate change into the clinical care setting (*p* < 0.001) and more willing to adapt their practice (*p* < 0.001) and to act as a role model (*p* < 0.001). Respondents with a right-wing political orientation tended to address climate change less with their patients (8.6% of right-wing respondents addressed the topic in more than a quarter of their consultations vs. 17.1% among left-wing respondents, *p* = 0.03) and were more likely to believe that addressing climate change was not their role (28% vs. 11% among left-wing respondents, *p* = 0.03).

These associations were found in adjusted multivariable models that included gender, age, political orientation (without/left wing/right wing) and place of practice (urban/semi-urban/rural). (Table 3).

## 5. Discussion

### 5.1. Concerns and Knowledge Regarding Climate and the Ecological Emergency

The idea that clinicians should play an active role in the climate crisis has been strongly advocated by medical editorialists [19,26,27,28]. The present study showed a concern about climate change and a willingness to get involved from a majority of clinicians in this sample of primary care physicians. Three in four respondents stated they were ready to adapt their practices, or at least to provide information on the relationship between climate change and health. However, we found that the majority of GPs felt uncomfortable advising patients on these themes. Looking into factors associated with the comfort and willingness of GPs to talk about climate change with patients, we found that GPs’ political affiliation and age played a role. Interestingly, it appears that currently, climate change is already a topic of discussion during consultations.

While our results confirm a real general concern of the sampled clinicians regarding climate change issues, they also reveal their limited knowledge of broad sustainability concepts such as planetary boundaries. Furthermore, although new concepts, such as “Planetary health”, are used increasingly in healthcare scholarly circles, clinicians are far from familiar with the term and the concept.

We can probably partly explain this by the broad framework of the concept and the difficultly in translating it to usable interventions at the local level, as well as perhaps the interdisciplinary nature of the planetary health perspective (ecosystem services, Earth system science and planetary boundaries) that has its roots in scientific disciplines generally considered distant from medical practice. It probably reflects to some extent the level of knowledge of the wider population.

There is a real need, therefore, for training on these issues. A large majority of respondents stated they were willing to learn more about climate change. In addition to GP-specific information and teaching for themselves and patients, a majority of respondents would be in favor of stronger outreach from the medical associations on climate change and health. Additionally, some researchers believe that health professionals and academics should be more proactive in public action towards the climate and ecological emergency [29]. Recognizing this important issue, the Swiss Medical Association adopted a promising strategy based on four axes: information, mitigation adaptation and role modeling. This strategy, adopted in October 2021, is aimed at promoting an adjustment and resilience of the health system in view of climate change [30].

### 5.2. Developing Interventions for the Clinical Setting

To date, very few studies have assessed if a planetary health perspective or any potential benefits between health and the planet could be integrated into the clinical care setting [31]. While our survey suggests that a high number of primary care clinicians are motivated to get involved with their patients regarding climate change and health, there is an important need to develop meaningful interventions and evaluate their outcomes.

Respondents seemed willing to work with the concept of co-benefits, suggesting this is a relevant direction in which to develop interventions. Among the possible co-benefit interventions proposed by the WONCA and others [32], diet, active mobility and contact with nature seemed well-accepted by the interviewed GPs, who were prepared to integrate them into their consultations [15]. Indeed, a more environmentally friendly diet (less processed food, less red meat and more vegetables), changing to active transport (walking or cycling instead of using polluting transport means) and exposure to green spaces have been long recognized as being beneficial to health [14,17,18]. Convincing evidence shows that choosing active transport and eating more vegetables and less meat allows for the reduction in personal carbon footprints. However, it is not clear how GPs could encourage their patients to change their behavior with this argument in mind, and there is a lack of data about which results could be expected. The acceptability and effectiveness of such a motion is likely to be highly influenced by structural conditions such as access to sustainable and healthy food (e.g., in terms of cost, distribution and cultural considerations), or the presence of safe and efficient cycling and walking infrastructure.

Not surprisingly, GPs were reluctant to discuss reproductive health with patients (to limit unwanted pregnancies and population growth). This is indeed a very sensitive and controversial topic, raising important ethical issues that have a long political history rather than health implications [33]. Energy choices or reducing personal environmental impact also seem to be difficult to address during a medical encounter.

Of note, most GPs agreed they had roles involving engagement with the community, for example, as health advocates. However, only a few of them had an active community role and even less reported being involved in politics. This implies that despite their growing concern about environmental degradation, it remains difficult for physicians, such as other sectors of the society, to realize the height of the step and the extent of the efforts that must be made to meet the IPCC’s warning call [34].

## 6. Strength and Limitations

The first limitation of this study is that we explored only a few elements related to climate change and did not consider all aspects of ecological degradation, such as biodiversity loss. On the other hand, we identified key questions regarding climate change that may be valuable to other environmental problems, such as the willingness to integrate climate change into clinical practice. Secondly, the study did not assess patient concerns and opinions on climate change, especially if they expected their doctor to address environmental topics during the medical consultation and trust their GP as a source of information about environmental issues. Thirdly, our study faced predictable selection bias as the topic might be perceived as sensitive and politically oriented. We expected GPs with more political involvement and strong views about climate change to be more likely to participate. As they were mainly based on a self-reported survey rather than observations, our results are also exposed to the risk of desirability bias, which may be limited by the anonymity of the survey [35]. Since we have no register of GPs containing the information necessary for a weighting of our sample, it is unfortunately not possible to correct this possible bias. Finally, although it revealed that climate change is sometimes addressed in medical consultation, the questionnaire gave no indication about the nature of this exchange, and, as such, does not allow differentiating small talk in which climate change might arise, from specific issues, concerns and fears about health and climate change brought up by the patient or the clinician. It nonetheless highlights how topical the subject has become.

One strength of the study is the representativeness of participants and the relatively large sample size (N = 497), even if the response rate was not very high (25%), as expected for such surveys [36]. The question on the political orientation showed that not only “green and left wing” GPs responded, as a quarter were right-wing, and another quarter did not mention their orientation. Even though we do not know the true right–left balance among GPs in the region from which our sample was drawn, according to discussions with different medical associations, the proportions seem rather characteristic. This observation mitigates the limitation of the response rate, reinforcing the accuracy of the results. Another strength of the study was exploring what we could do in the clinical setting (i.e., co-benefit interventions, information on climate change for patients).

## 7. Conclusions

This study revealed a sort of paradoxical gap between the strong willingness of GPs to integrate the impact of climate change on health into their clinical activities and their lack of climate knowledge, adequate interventional ideas and scientific evidence on how the two interact. A promising way forward may be to develop co-benefit interventions adapted to the clinical setting focused on diet, active mobility and connecting with nature at the local level.

## Figures and Tables

**Figure 1 ijerph-19-04901-f001:**
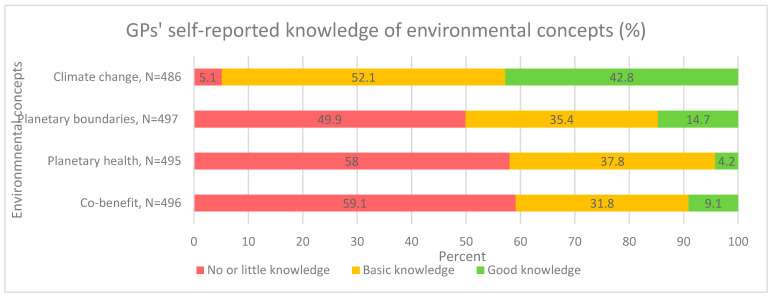
General practitioners’ self-reported knowledge of environmental concepts (%).

**Figure 2 ijerph-19-04901-f002:**
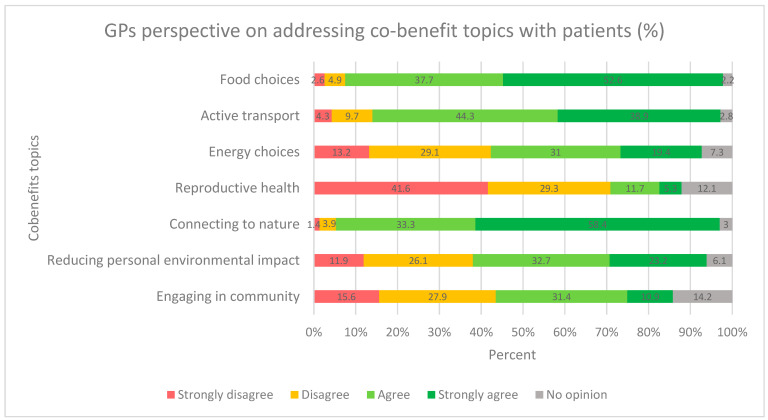
Willingness of respondents to address co-benefit topics from the WONCA declaration with patients. (%) N = 497.

**Table 1 ijerph-19-04901-t001:** Demographic characteristics of respondents, N (%). Not all categories represent the total number of respondents (N = 497) due to missing responses.

	N (%)	Total of Respondents N
Age (y) mean ± SD	52 ± 11.4	493
30–45	167 (33, 9)	
46–60	207 (42)	
>60	116 (24, 1)	
Gender		
Female	261 (52, 7)	495
Male	234 (47, 3)	
Years in practice (y) mean ± SD	25 ± 11	484
Primary work site		
Urban	284 (57, 4)	495
Semi-urban	124 (25)	
Rural	87 (17, 6)	
Type of office		
Alone	155 (31, 6)	491
Group office	336 (68, 4)	
Primary work setting		
Private	471 (96, 1)	491
Public	19 (3, 9)	
Active in associative activity		
Private	98 (20)	490
Professional	32 (6, 6)	487
Political affiliation		
Right wing/Conservative	106 (21, 3)	432
Left wing/Liberal	172 (34, 6)	
Other political affiliation	11 (2, 2)	
No political affiliation	143 (28, 8)	

**Table 2 ijerph-19-04901-t002:** GPs’ opinion on adapting the clinical setting to health impact of climate change and providing information about the issue to patients, N (%).

	Strongly Agree	Agree	Disagree	Strongly Disagree	No Opinion	N
Clinical practice should be adapted to health impacts of climate change	100 (20.1)	295 (59.4)	52 (10.5)	14 (2.8)	36 (7.2)	497
GPs have a role in providing information to patients about climate change and its links to human health	187 (37.6)	224 (45.1)	46 (9.3)	20 (4)	20 (4)	497
I feel comfortable advising patients about the health impact of climate change	35 (7.1)	169 (34.1)	194 (39.1)	62 (12.5)	36 (7.3)	496

**Table 3 ijerph-19-04901-t003:** Odds ratio of the multivariable logistic regression.

Variables	Feels Comfortable Addressing Climate Change with Patients		Willingness to Integrate the Theme of Climate Change into the Clinical Care Setting		Willingness to Adapt the Clinical Setting to Climate Change Consequences		Willingness to Be a Model Role for the Population in Terms of Sustainability		Willingness to Provide Patients Information about Health Impact of Climate Change		Climate Change Topic Addressed in Consultation	
	OR (95% CI)	*p*-Value	OR (95% CI)	*p*-Value	OR (95% CI)	*p*-Value	OR (95% CI)	*p*-Value	OR (95% CI)	*p*-Value	OR (95% CI)	*p*-Value
Age (by 5 years)	1.05 (1.03–1.07)	<0.01	1.04 (1.02–1.07)	<0.01	1.01 (0.98–1.04)	0.50	1.03 (1.01–1.06)	0.01	1.04 (1.01–1.07)	0.01	1.07 (1.04–1.1)	<0.01
Sex (Reference = women)	0.81 (0.52–1.27)	0.36	1.31 (0.77–2.23)	0.32	1.21 (0.65–2.28)	0.55	1.13 (0.63–2.04)	0.69	1.64 (0.88–3.13)	0.13	1.62 (0.87–3.05)	0.13
Place of practice (Reference = Urban)												
Semi-urban	1.21 (0.73–2.02)	0.46	1.25 (0.69–2.32)	0.47	1.34 (0.68–2.80)	0.42	1.1 (0.58–2.15)	0.77	1.10 (0.58–2.24)	0.79	0.87 (0.43–1.71)	0.7
Rural	1.14 (0.64–2.02)	0.65	0.76 (0.4–1.48)	0.41	1.36	0.47	1.74 (0.79–4.29)	0.19	1.74 (0.79–2.99)	0.58	0.97 (0.43–2.03)	0.93
Political orientation (Reference = Without)												
Left wing	1.94 (1.19–3.20)	<0.01	2.34 (1.29–4.30)	<0.01	2.93 (1.47–6.06)	<0.01	3.66 (1.79–7.87)	<0.001	3.66 (1.79–5.90)	0.009	1.15 (0.62–2.17)	0.65
Right wing	0.94 (0.53–1.66)	0.83	0.66 (0.36–1.19)	0.17	1.52 (0.76–3.14)	0.25	0.67 (0.36–1.25)	0.21	0.67 (0.36–1.40)	0.33	0.4 (0.16–0.89)	0.03

## Data Availability

The datasets analyzed during the current study are available from the corresponding author on reasonable request.

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
