# Peer review of "Talking about Climate Change and Environmental Degradation with Patients in Primary Care: A Cross-Sectional Survey on Knowledge, Potential Domains of Action and Points of View of General Practitioners"

_ijerph, 2022, doi:10.3390/ijerph19084901_

Round 1

Reviewer 1 Report

Reviewer comments

Manuscript ID: ijerph-1637165

General comments: The medical community, in particular clinicians, have the potential to play a leading role in raising awareness about the health effects of the climate crisis, while also advocating for more sustainable health systems. The goal of this survey-based analysis was to examine general practitioner’s knowledge and perspectives about the health impacts of climate change in Switzerland. This study focuses on “GPs as they are the first point of contact of healthcare and arguably potential leaders to draw attention of communities to climate change and to advocate change at the policy level due to their proximity to the community they serve.” This is an interesting analysis that sheds light on the willingness of GPs to get involved in the climate crisis conversation with their patients and how equipped this group feels in advising their patient population on climate-health concerns. This study will be of interest to readers, clinicians, and the public health community more broadly. I hope to see more studies like this one to better understand where GPs and the medical community, more broadly, stand on this issue and to better understand the education, tools, and resources this group needs to be better local and policy advocates, as well as provide better care for the first and worst impacted in their community in response to a changing climate.

INTRODUCTION

The authors mention that “Clinical practice is already and will be further affected by 47 climate change and thus clinicians will have to anticipate and adapt healthcare to the bur- 48 den of diseases induced by climate change. 5,6”Please elaborate on with illustrative examples

Can the authors better define what is meant by “ecosystem functioning” and “planetary boundaries” for this work?

Minor: This includes for e.g. active mobility [change e.g., to for example]

METHODS

“Finally, we used multiple linear and logistic regression models to estimate the association between variables of interest and the main demographic characteristics (age, gender, place of practice and political orientation).” Can the authors please provide more details on the primary outcomes of interests, covariates, etc?

One question – why this group of GPs and why now? What motivated the survey?

Minor: so they were able to progress through the 114 questionnaire. (add to)

Re: interventions - Why not also look at the willingness of health systems or clinical practices to change to more sustainable practices?

Will the authors publish the survey in the supporting materials for broader dissemination?

RESULTS

Table 5 – Why not look at age as categorical to show distinction between younger vs older responses (e.g., <39, 40-64, 65 years+)?

DISCUSSION

One important note in contextualizing findings is that results are only generalizable to this sample of GPs.

The authors note: “It implies that despite their growing concern about environmental degradation, several mechanisms, such as psychological barriers or not realizing the full implication of the threats, prevent any current action to help meet the IPCC’s warning call.31” The transition here from physicians being engaged in their community to psychological barriers or a lack of perception of risk concerning the climate crisis and public health is abrupt. Can the author team better hone in on the connection they are trying to make here?

Strengths and limitations

“On the other hand, we have the impression we identified key questions regarding climate change that may be valuable to other environmental problems.” A citation might be more relevant here to replace “we have the impression we identified”.

“One strength of the study is the representativeness of participants and the relatively 326 large sample size (N=497), even if the response rate was not very high (25%) but rather 327 good for such surveys” Please add citation here

Reviewer 2 Report

The aim of this study was to assess GPs knowledge and perspectives about the health impacts of climate change. The study was carried out in Switzerland using a questionnaire. Although there has been screens of physician’s views on this topic previously, none have been based in this country, and none have specifically screened for the views on co-benefit action explicitly. Therefore, this paper is an important contribution to the literature.

Line- 27: Most of ‘the’ GP’s

Lines 67-70: It notes here, “a number of papers proposed”, but then there are no formal references to these papers at the end of the sentence. It would be good to reference some of these papers with this sentence for those that want to learn more, but also backs up your statement that there are papers available in this regard.

Lines-70-71: Same as above, this sentence notes “repeatedly highlighted”. It could be good to at least add one reference to this sentence to back up the statement.

Lines-139-148-How did you account for the response bias inherent in survey tools with the convenience sample used in the statistical analysis (i.e., making inferences in a non-random sample)?

-Lines 270-273- Could use a slight grammar adjustment for better flow of sentence.

-Lines 275-277- Could use a slight grammar adjustment for better flow of sentence.

-Lines-289-290- End of sentence could use a slight grammar adjustment for better flow of sentence I.e., allows for the reduction of personal...)

-Lines 290-291- “and few solid results” seems a little out of place. Sentence could use some adjustments for better flow and clarity.

-Lines 327-328- “but rather good for surveys” also seems a little out of place how it is currently written. Sentence could use some adjustments for better flow and clarity.

-Line 329- …”responded as a quarter were right…” (need to add word ‘a’).

-Lines-343-344- In many countries even if no personal data on health are collected, it still requires a more formalized exemption. Did a specific ethics body review the work and confirm that this project was exempt?
